# Investigating autism associated genes in *C. elegans* reveals candidates with a role in social behaviour

**Helena Rawsthorne**, **Fernando Calahorro**, **Lindy Holden-Dye**, **Vincent O' Connor**, **James Dillon** *

School of Biological Sciences, Highfield Campus, University of Southampton, Southampton, United Kingdom

* j.c.dillon@soton.ac.uk

**Data Availability Statement:** All relevant data are within the manuscript and its Supporting Information files.

## Abstract

Autism spectrum disorder (ASD) is a neurodevelopmental disorder characterised by a triad of behavioural impairments and includes disruption in social behaviour. ASD has a clear genetic underpinning and hundreds of genes are implicated in its aetiology. However, how single penetrant genes disrupt activity of neural circuits which lead to affected behaviours is only beginning to be understood and less is known about how low penetrant genes interact to disrupt emergent behaviours. Investigations are well served by experimental approaches that allow tractable investigation of the underpinning genetic basis of circuits that control behaviours that operate in the biological domains that are neuro-atypical in autism. The model organism *C. elegans* provides an experimental platform to investigate the effect of genetic mutations on behavioural outputs including those that impact social biology. Here we use progeny-derived social cues that modulate *C. elegans* food leaving to assay genetic determinants of social behaviour. We used the SAFRI Gene database to identify *C. elegans* orthologues of human ASD associated genes. We identified a number of mutants that displayed selective deficits in response to progeny. The genetic determinants of this complex social behaviour highlight the important contribution of synaptopathy and implicates genes within cell signalling, epigenetics and phospholipid metabolism functional domains. The approach overlaps with a growing number of studies that investigate potential molecular determinants of autism in *C. elegans*. However, our use of a complex, sensory integrative, emergent behaviour provides routes to enrich new or underexplored biology with the identification of novel candidate genes with a definable role in social behaviour.

## Introduction

Autism spectrum disorder (ASD) is a pervasive neurodevelopmental behavioural disorder. ASD is characterised by a triad of behavioural impairments, these being repetitive behaviours and impairment to verbal and social communication [1]. Neuro-atypical individuals have been shown to produce altered behavioural outputs in response to a range of sensory cues [2], including chemosensory cues that drive social behaviours [3]. Impairment within the

**Funding:** HR was funded by the The Gerald Kerkut Charitable Trust, http://www.kerkut-trust.org.uk. The funders had no role in study design, data collection and analysis, decision to publish, or preparation of the manuscript.

**Competing interests:** The authors have declared that no competing interests exist.

integration of sensory stimuli is thought to underlie the altered perception of such cues [4]. This highlights the importance of neural circuits in the processing of sensory information to coordinate a behavioural output in the social domain [5].

It is well established that there is a strong genetic contribution in autism [6]. The genetic architecture of ASD is complex with hundreds of genes of varying penetrance implicated in its aetiology [7]. This is complicated further by the interplay between genetic variants in the form of rare, highly penetrant, and common low penetrant variants [8,9]. Common variants attribute polygenic risk in ASD with mutations to multiple loci having additive effects on a given phenotype [8]. The burden of common variants in an individual's genetic background can influence the degree of risk a rare variant can impose [9]. The combinatorial effect of rare and common variants contributes to the spectrum of phenotypes displayed across autism cases [10].

ASD associated genes span across a range of biological functions, for example synaptic, cell signalling and epigenetic modification [11]. Evidence suggests that although ASD genes are functionally diverse they are connected through protein interaction networks [12] and control processes such as neuronal morphology and synaptic function that modulate the activity state of neural networks [8,13,14]. This means that the consequence of even a single genetic variant can be widespread through inter-connecting gene networks and have emergent effects on neural circuits [8]. For many ASD associated genes it is still unclear how they function in neural networks which underpin behavioural phenotypes [15], such as disrupted social and motor behaviour. However, investigating determinants of defined neural circuits underpinning autism associated neuro-atypical behaviour is providing traction for discrete investigation of complex traits. Study of distinct behaviours in mice has begun to unpick the effect of genetic disruption on molecular circuits and synapse function [16–18]. Additionally, the impact of genetic variation on a number of synaptic genes has been extensively studied in various animal models [19–24].

Animal models highlight the value of using orthologues to understand the function of ASD associated genes in behavioural domains associated with autism [25–28]. *C. elegans* provides a tractable system that allows for the high throughput of genetic determinants to be investigated in a simple nervous system [29]. Conservation of genes involved in synapse function and the use of integrative neurons highlights the utility of *C. elegans* neuronal function and how it coordinates complex sensory integrative behaviours that model the disruptions that are expressed through genetic mutations associated with autism [30,31]. The genetic homology between the *C. elegans* and mammalian genome [32], and the fact that mutant strains are widely accessible, means that the *C. elegans* model lends itself to the systems level analysis of disease associated genes. This has led to a plethora of studies using single gene analysis to investigate the impact of genetic mutation to ASD associated gene orthologues on behavioural output [33]. As well as this, *C. elegans* have been utilised in multiple high-throughput screens which have largely used morphological and locomotory readouts to screen for behavioural deficit [34–36].

Behavioural output in response to integration of sensory cues can be assayed in *C. elegans* by way of food leaving behaviour. The propensity of a worm to leave a lawn of bacterial food can be modulated by multiple sensory cues [37]. It has been shown that in the presence of increasing numbers of progeny, adult worms will leave an otherwise replete food lawn in a dose-dependent manor. This progeny-dependent food leaving behaviour is the result of inter-organismal communication and is thought to be underpinned by a novel social circuit [38]. The utility of this social paradigm to probe autism related dysfunction was demonstrated by showing that when a penetrant mutation of human neuroligin is introduced into the worm orthologue, *nlg-1*, it results in disrupted progeny induced food leaving behaviour [39].

We have used this bona fide social paradigm to investigate genetic determinants associated with human ASD. Investigation of *C. elegans* orthologues in a subset of candidate genes identified a number that disrupt a social behavioural paradigm in the worm. Furthermore, we show that whilst a large proportion of mutants displayed behavioural deficit in the social domain, there was limited disruption to the other phenotypes investigated suggesting a selective behavioural deficit. Identification of novel candidate genes in this way has also highlighted key biological functional domains that appear to play an important role in social behaviour, therefore shedding light on the functional contribution ASD associated genes may have on the disrupted phenotypes associated with this disorder.

## Materials and methods

### Prioritising ASD associated genes for study in *C. elegans*

To identify genes associated with ASD we used SFARI Gene Archive (http://gene-archive.sfari.org/, version 3.0). Within this database the Human Gene Module (https://gene-archive.sfari.org/database/human-gene/) ranks genes from 1–6 based on the evidence supporting the gene's association with ASD. Genes in category 1-High confidence and category 2-Strong candidate were selected for analysis due to the fact that data implicating those genes in ASD reach genome wide significance and there is evidence for the variant having a functional effect in humans. Orthologous genes in *C. elegans* were identified by searching the human gene name in WormBase (https://wormbase.org/, version WS264) and using the human gene Ensembl ID in OrthoList (http://www.greenwaldlab.org/ortholist/). *C. elegans* strains available for order from the *Caenorhabditis* Genetics Centre (CGC) and/or the National BioResource Project (NBRP) were prioritised for investigation. Using information gathered from WormBase, CGC, NBRP and a literature review, mutants were excluded if they were lethal, sterile or uncoordinated. Thus, we filtered for candidates best suited to investigation in the food lawn based assay. The prioritised *C. elegans* mutant strains for study can be found in Table 1. Genes were ascribed to one of five functional categories: synaptic, neuronal, cell signalling, epigenetic modifiers and phospholipid metabolism based on their function described by UniProtKB (https://www.uniprot.org/uniprot/). Genes described as having a role in synaptic transmission, structure, activity or plasticity were categorised as 'synaptic'. Genes with a role in neuronal excitability or adhesion were categorised as 'neuronal'. Genes described as having a predominant role in cell signalling pathways were categorised as 'cell signalling'. Genes with a role in transcriptional regulation or chromatin remodelling were categorised as 'epigenetic modifier'. The gene MBOAT7 is described as functioning in phospholipid metabolism and so was categorised as 'phospholipid metabolism'.

### *C. elegans* culturing and strains used

All *C. elegans* strains were maintained using standard conditions [80]. *C. elegans* were age synchronised by picking L4 hermaphrodites onto a new plate 18 hours prior to behavioural assays. Bristol N2 were used as wild-type control. All other strains used can be found in Table 1. Strains were obtained from either the *Caenorhabditis* Genetics Center or National BioResource Project.

### Food leaving assay

5cm NGM (nematode growth medium) plates were prepared using a standard protocol [80]. 50μl of OP50 *E.coli* at $OD_{600}$ of 0.8 was gently spotted on the middle of an unseeded plate. Approximately 18 hours following this, seven L4+1 day old hermaphrodites were picked onto

**Table 1. Summary of human genes prioritised for study in *C. elegans* mutant strains.**

| Human gene | Gene name | Protein function | *C. elegans* orthologue | Allele | Strain name | Out-crossed | Mutation | Mutation effect | Behavioural phenotype | Gene expression |
|---|---|---|---|---|---|---|---|---|---|---|
| **Synaptic** | | | | | | | | | | |
| GRIA1 | Glutamate ionotropic receptor AMPA type subunit 1 | Glutamate receptor | *glr-1* | *n2461* | KP4 | 4 | Nonsense mutation in codon 807 [40] | LOF [40] | Defective local search behaviour [41] | AVA,AVB, AVD,AVE, PVC,AIB, RMD, RIM,SMD, AVG PVQ,URY, RIS, AVJ, DVC, RME, RIG [40] |
| | | | *glr-2* | *tm669* | FX00669 | 0 | Complex substitution [42] | Unpublished | Enhanced gustatory plasticity [43] | AVA,AVD, AVE PVC,RMDV, RMDD,AIA, AIB,AVG, RIG, RIA,M1 [44] |
| | | | *glr-2* | *ok2342* | RB1808 | 0 | Deletion [42] | Unpublished | Unknown | |
| GRIN2B | Glutamate Ionotropic Receptor NMDA Type Subunit 2B | NMDA receptor | *nmr-2* | *ok3324* | VC2623 | 1 | Deletion [42] | LOF [34] | Reduced swimming locomotion [34] | AVA,AVD, AVE, RIM,AVG, PVC [44] |
| | | | *nmr-2* | *tm3785* | FX03785 | 0 | Deletion [42] | LOF [45] | Impaired learning [46] | |
| NLGN3 | Neuroligin 3 | Synaptic adhesion | *nlg-1* | *ok259* | VC228 | 6 | Deletion to half of cholinesterase-like domain and TMD [22] | Null [22] | Reduced spontaneous reversals [22] Reduced pharyngeal pumping [47] | VA,DA,AIY, URB,URA, PVD,HSN, ADE,URX, AVJ, ALA [22,47] |
| NRXN1 | Neurexin 1 | Synaptic adhesion | *nrx-1* | *ds1* | SG1 | 3 | Deletion in the long *nrx-1* isoform [48] | Unpublished | Unknown | Nervous system, GABAergic neurons [49,50] |
| | | | *nrx-1* | *tm1961* | FX01961 | 0 | Deletion in the long *nrx-1* isoform [48] | Truncated protein [24] | Deficient gentle touch response [24] | |
| PTCHD1 | Patched domain containing 1 | Synaptic receptor | *ptr-5* | *gk472* | VC1067 | 0 | Deletion [42] | Unpublished | Unknown | Unknown |
| SHANK 2/3 | SH3 and multiple ankyrin repeat domains | Synaptic scaffold | *shn-1* | *ok1241* | RB1196 | 0 | Deletion covering PDZ domain and proline rich motif [51] | LOF [51] | None reported [51] | Widely expressed [52] |
| | | | *shn-1* | *gk181* | VC376 | 0 | Deletion covering most of ANK repeat and entire PDZ domain [51] | LOF [51] | Unknown | |
| SLC6A1 | Solute carrier family 6 member 1 | GABA transporter | *snf-11* | *ok156* | RM2710 | 6 | Deletion [53] | Putative null [53] | None reported [53] | AVL,RIBR, ALA, RIBL,GLRV, RME,AVF, EF1, EF2,EF3,EF4, Body wall muscle [54] |
| | | | *snf-11* | *tm625* | FX00625 | 0 | Deletion and insertion [53] | Putative null [53] | Unknown | |

*(Continued)*

**Table 1.** (Continued)

| Human gene | Gene name | Protein function | *C. elegans* orthologue | Allele | Strain name | Out-crossed | Mutation | Mutation effect | Behavioural phenotype | Gene expression |
|---|---|---|---|---|---|---|---|---|---|---|
| SYNGAP1 | Synaptic Ras GTPase activating protein 1 | Ras GTPase activating protein | *gap-2* | *tm748* | JN147 | 0 | Complex substitution [42] | LOF [55] | No effect on body bends [55] | Widely expressed [55] |
| | | | *gap-2* | *ok1001* | VC680 | 0 | Complex substitution [42] | Unpublished | Unknown | |
| **Neuronal** | | | | | | | | | | |
| CACNA1H | Calcium voltage-gated channel subunit Alpha1 H | Calcium channel | *cca-1* | *ad1650* | JD21 | 7 | Deletion [56] | LOF [56] | Reduced pharyngeal pumping [56] | Pharyngeal muscle, neurons in pharynx and VNC [56] |
| CNTN4 | Contactin 4 | Axonal adhesion | *rig-6* | *ok1589* | VC1125 | 0 | Deletion [57] | Hypo-morphic [57] | None reported [57] | Widely expressed in nervous system [58] |
| | | | *rig-6* | *gk376* | VC884 | 0 | Deletion-knocks down expression of isoform a only [57] | Hypo-morphic [57] | Unknown | |
| **Cell signalling** | | | | | | | | | | |
| DYRK1A | Dual specificity tyrosine phosphorylation regulated kinase 1A | Protein kinase | *hpk-1* | *pk1393* | EK273 | 6 | Deletion to most of kinase domain [59] | Null [59] | Reduced lifespan [59] | Gonad, nervous system–not otherwise specified [59] |
| | | | *mbk-1* | *pk1389* | EK228 | 6 | Deletion to most of kinase domain [60] | Putative null [60] | Reduced lifespan [60] | Somatic tissue, not otherwise specified [61] |
| | | | *mbk-1* | *ok402* | RB677 | 0 | Unknown | Unknown | Reduced swimming locomotion [34] | |
| PTEN | Phosphatase and tensin homolog | Protein phosphatase | *daf-18* | *e1375* | CB1375 | 0 | Insertion [62] | Reduction of function [62] | Chemotaxis deficit [63] | Widely expressed [64] |
| | | | *daf-18* | *ok480* | RB712 | 0 | Deletion [65] | Putative null [65] | Abnormal mitotic arrest in dauer [66] | |
| **Epigenetic Modifiers** | | | | | | | | | | |
| CHD8 | Chromodomain helicase DNA binding protein 8 | Transcription factor | *chd-7* | *gk290* | VC606 | 0 | Deletion [42] | Unpublished | Reduced swimming locomotion [34] | Unknown |
| | | | *chd-7* | *gk306* | VC676 | 0 | Deletion [42] | Unpublished | Impaired habituation [36] | |
| FOXP1 | Forkhead box P1 | Transcription factor | *fkh-7* | *gk793* | VC1646 | 0 | Deletion [42] | Unpublished | Unknown | Widely expressed [67] |
| IRF2BPL | Interferon regulatory factor 2 binding protein like | Transcription factor | *tag-260* | *ok1339* | VC812 | 0 | Insertion [42] | Putative null [68] | Unknown | Unknown |
| KDM6A | Lysine-specific demethylase 6A | Histone demethylase | *jmjd-3.1* | *gk387* | VC912 | 0 | Deletion [42] | Unpublished | Unknown | PDA motor neuron and Y cell [69] |
| | | | *jmjd-3.1* | *gk384* | VC936 | 0 | Insertion [42] | Null [70] | Unknown | |
| KMT5B | Lysine methyltransferase 5B | Transcription factor | *set-4* | *n4600* | MT14911 | 2 | Deletion [71] | LOF [71] | Deficient dauer arrest [71] | Nervous system, not otherwise specified [71] |
| | | | *set-4* | *ok1481* | VC997 | 0 | Deletion [71] | LOF [71] | Deficient dauer arrest [71] | |

(*Continued*)

**Table 1.** (Continued)

| Human gene | Gene name | Protein function | *C. elegans* orthologue | Allele | Strain name | Out-crossed | Mutation | Mutation effect | Behavioural phenotype | Gene expression |
|---|---|---|---|---|---|---|---|---|---|---|
| SETD2 | Histone-lysine N-methyltransferase SETD2 | Transcriptional regulation | *met-1* | *n4337* | MT16973 | 4 | Deletion [72] | LOF [73] | Sterility at 25°C [72] | Broadly expressed [74] |
| | | | *met-1* | *tm1738* | FX01738 | 0 | Deletion [72] | Putative null [72] | Sterility at 25°C [72] | |
| SETD5 | Histone-lysine N-methyltransferase SETD5 | Transcriptional regulation | *set-26* | *tm3526* | FX03526 | 0 | Deletion [42] | Putative null [75] | None reported [75] | Widely expressed [76] |
| | | | *set-24* | *n4909* | MT16133 | 0 | Unknown [42] | Unknown | None reported [77] | Germline specific [74] |
| | | | *set-9* | *n4949* | MT16426 | 1 | Deletion [42] | Putative null [75] | None reported [77] | Germline specific [76] |
| **Phospholipid metabolism** | | | | | | | | | | |
| MBOAT7 | Membrane bound O-acyltransferase domain containing 7 | Acetyl transferase | *mboa-7* | *ok1028* | RB1071 | 0 | Deletion [78] | LOF [78] | Unknown | Muscle, vulva, intestine [79] |
| | | | *mboa-7* | *gk399* | VC942 | 0 | Deletion [78] | LOF [78] | Egg laying deficit [79] | |
| | | | *mboa-7* | *tm3536* | FX03536 | 0 | Deletion [78] | LOF [78] | Developmental defects [78] | |
| | | | *mboa-7* | *tm3645* | FX03645 | 0 | Deletion [78] | LOF [78] | Unknown | |

Human genes were used to ascribe functional domains. For each human gene the *C. elegans* orthologue used for investigation is listed and the mutant allele, known phenotypes and expression of the gene indicated. LOF stands for loss of function. AA stands for amino acids. TMD stands for transmembrane domain. LNS stands for laminin-neurexin-sex hormone-binding globulin. References are indicated.

the centre of the bacterial lawn. Plates were then incubated at 20°C for 24 hours, during this time the seven assay worms lay eggs which produces a progeny laden lawn. In all food leaving assays the number of food leaving events were counted manually during a 30 minute observation period using a binocular dissecting microscope (Nikon SMZ800; X10). A food leaving event was defined as when the whole of the worm's body came off the bacterial food lawn, as previously described [39]. Following each food leaving assay the % proportion of eggs, L1 and L2 progeny on the plate was calculated. For all food leaving assays N2 and *nlg-1(ok259)* animals were analysed in parallel to other mutant cohorts. Investigators were blind to the genotypes being observed.

## Pre-conditioned food leaving assay

NGM plates were prepared and seeded as described above. 18 hours after seeding assay plates, half were pre-conditioned with progeny using the protocol described previously [38] and the remaining plates were used as matched unconditioned controls. In the preconditioned plates 10 gravid adults were picked onto the centre of the bacterial lawn and left to lay 150–200 eggs before being picked off. 18 hours after this, for each mutant under investigation, seven L4+1 day old hermaphrodites were picked onto the centre of a naïve bacterial food lawn. This acts as a matched unconditioned control. Another seven L4+1 day old hermaphrodites were picked onto a pre-conditioned bacterial food lawn in which 150–200 eggs had developed for 18 hours. The plates were then incubated at 20°C for 2 hours, this time course and pre-conditioning is equivalent to a 24 hour naive incubation. Leaving events were then observed for 30 minutes as described above.

## Pharyngeal pumping

Following the measurement of food leaving at the 24 hour time point, feeding behaviour was quantified by counting the pharyngeal pumping of three of the seven worms. The worms selected for these measurements were on food for the observation period. One pharyngeal pump was defined as one cycle of contraction-relaxation of the terminal bulb of the pharyngeal muscle. This behaviour was measured for 1 minute using a binocular dissecting microscope (Nikon SMZ800; X63) and the pumps per minute for each worm recorded from a single observation [81].

## Thrashing

Thrashing analysis was performed on the *C. elegans* mutants that were investigated in the pre-conditioned food leaving assay. Using a 24 well plate, 6–7 N2 or mutant worms were picked per well containing 500μl of M9 with 0.1% bovine serum albumin and left for 5–10 minutes before thrashing was observed. For each worm thrashing was counted for 30 seconds. Each thrash was defined as a complete movement through the midpoint of the worms body and back. For each mutant under investigation N2 control worms were analysed in parallel and at least two separate assays were performed. Investigators were blind to the genotypes being investigated.

## Statistical analysis

Statistical analysis was performed using GraphPad Prism 8 software. Data are expressed as mean or mean ±SEM as indicated in the figure legend. Statistical tests and post-hoc analysis is indicated in the figure legends. Significance level was set to $P < 0.05$.

## Results

### Selection of human ASD associated genes for study using *C. elegans* social behaviour

The genetic architecture of autism is complex with over 1,000 genes currently implicated in the disorder [11]. Furthermore, the functional contribution that many of these genes make to the behavioural domains implicated in ASD remains unclear. We have created a pipeline (Fig 1) to select *C. elegans* orthologues of human ASD associated genes and that can be investigated in a paradigm of social behaviour in the worm.

We used SFARI Gene, a growing database which categorises ASD risk genes based on the strength of evidence supporting the association. We prioritised 91 genes ranked by SFARI Gene Archive (accessed October 2018) as category 1-high confidence and category 2-strong candidate. Of these 91 genes, 84% (76/91) had at least one orthologue in *C. elegans*. A mutant strain was available for 84% (64/76) of the orthologous genes using the criteria that the mutant strain was available from the CGC and/or NBRP. Of these, 43 genes had available mutants that were either lethal, sterile or uncoordinated (Fig 1). We considered that such phenotypes rendered these mutants unsuitable for investigation in the social behaviour assay. On this basis we selected 40 *C. elegans* mutants spanning 21 human ASD associated genes for further investigation (Table 1). The human ASD associated genes were each assigned to a group based upon the functional description of the encoded protein in UniProtKB database. The functional groupings were: synaptic, neuronal, cell signalling, epigenetic modifiers and phospholipid metabolism. This led to a distribution of candidates highlighting 43% as synaptic genes, 33% as epigenetic modifiers, 10% as cell signalling, 9% neuronal and 5% phospholipid metabolism (Fig 1).

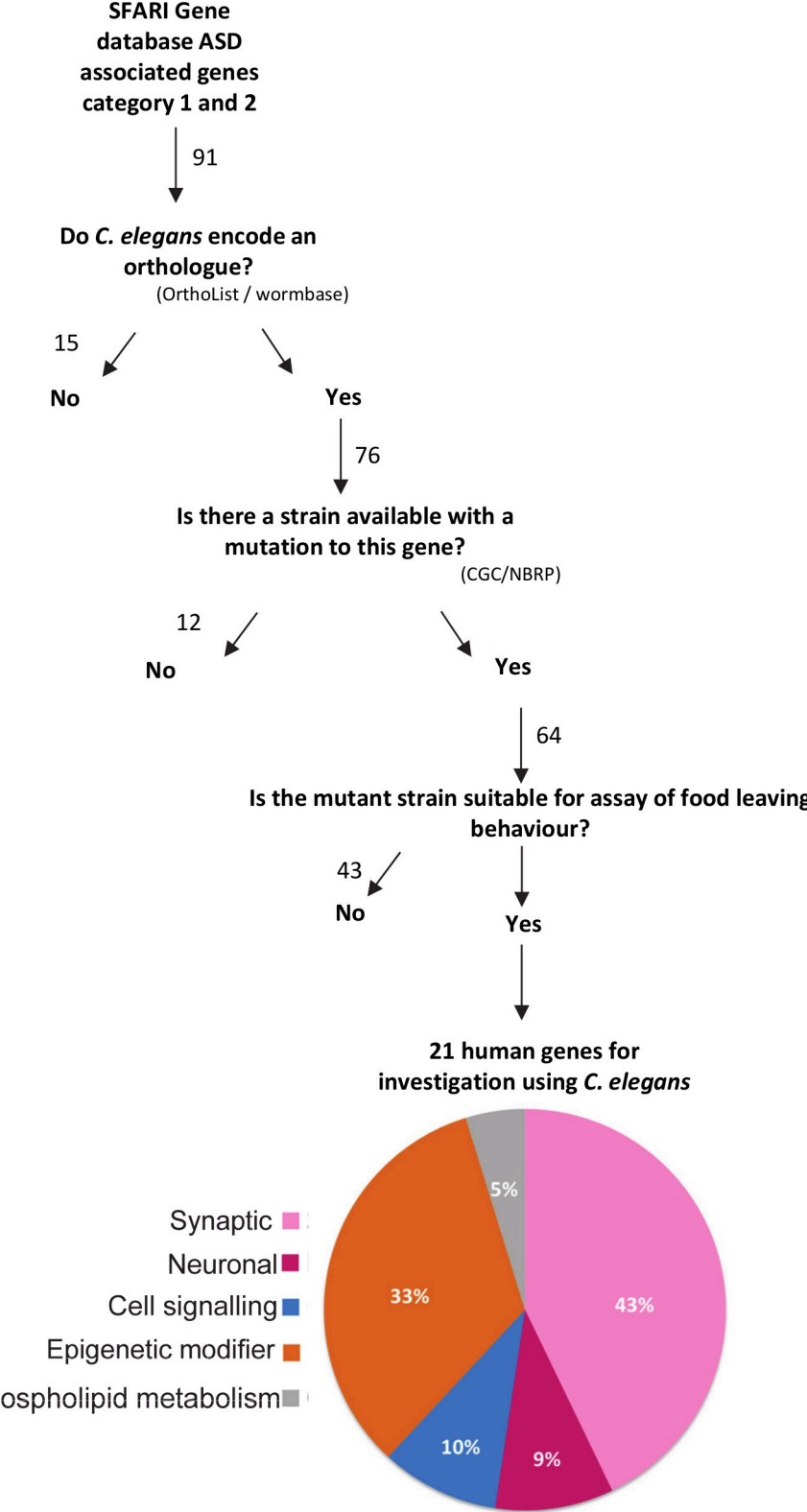

**Fig 1. Prioritisation and categorization of the *C. elegans* orthologues of prioritised human ASD associated genes.** High confidence ASD associated genes in category 1 and 2 in SFARI Gene Archive were input. The pipeline selects human genes which have an orthologue in *C. elegans* which can be studied in an available mutant strain which is neither lethal, sterile or uncoordinated. In brackets are the resources used for analysis. CGC–*Caenorhabditis* Genetics

Center. NBRP–National BioResource Project. The number of genes analysed using SFARI Gene Archive (https://gene-archive.sfari.org/, accessed October 2018) are stated. The pie chart indicates the percent of the 21 human genes that were placed into five functional groupings.

## Screening mutants using food leaving behaviour identifies ASD associated genetic determinants of social behaviour

To investigate food leaving behaviour, mutants were picked onto the centre of a bacterial lawn and food leaving events were measured after 24 hours. During the 24 hour incubation period the adult worms lay eggs which hatch into *C. elegans* progeny. It has been previously shown that progeny-derived social cues mediate a progeny-dependent increase in adult food leaving behaviour [38]. In accordance with previous findings we observed that N2 worms left the food lawn after 24 hours at a rate of approximately 0.088 leaving events/worm/minute (Fig 2). We had previously established a blunted food leaving response for the *nlg-1(ok259)* mutant [39] and this was used as an internal measure in the current assays (Fig 2).

Against this backdrop, N2 and *nlg-1(ok259)* were investigated alongside the selected mutants we filtered through following initial selection from the SAFRI Gene data base. This comparison showed that 23 of the 39 *C. elegans* mutants showed a mean food leaving rate lower than that of *nlg-1(ok259)* suggesting food leaving impairment (Fig 2). Mutants with a reduced food leaving phenotype were distributed across the five functional categories we defined suggesting genetic disruption within a range of molecular determinants from distinct biological domains may contribute to the emergence of *C. elegans* social behaviour.

As part of the investigation, where possible, we analysed two or more mutant alleles for a single gene (Fig 2). For some mutants, for example for *gap-2* and *rig-6* mutants, the two mutants phenocopied one another and showed a food leaving rate similar to that of N2. Interestingly, we found two loss-of-function *nmr-2* mutants which also phenocopied one another but showed significant food leaving impairment (Fig 2). In contrast, there were also instances where mutant alleles did not phenocopy each other. For example *nrx-1* and *chd-7* mutants showed one mutant allele with impaired food leaving and one with a behavioural response to progeny similar to N2 (Fig 2).

## Impaired social behaviour of mutants is likely a selective response to progeny derived social cues

Previous work has identified the value of investigating additional behaviours that can be scored in the observational arena [36]. In this respect the food leaving assay allows for multi-tracking phenotypic analysis including pharyngeal pumping, development and egg laying. In the case of pharyngeal pumping and egg laying, this reflects the output of a defined neuromodulation and the possible consequence progeny exposure might have on this. In the case of development, this provides insight into whether the mutations perturb gross development, a useful consideration in a neurodevelopmental disorder.

After each food leaving assay we quantified the pharyngeal pump rate of the mutants. Pharyngeal pumping is modulated via external sensory cues such as food [82]. Therefore we wanted to test whether another sensory regulated behaviour was affected in these mutants. 87% of mutants showed no pumping phenotype (Fig 3). In fact the majority of mutants with impaired social behaviour (Fig 2) had a pumping rate similar to N2 (Fig 3). This shows that most mutants with reduced food leaving behaviour are largely capable of responding to food-dependent sensory cues and co-ordinating normal feeding behaviour. In addition, the *cca-1 (ad1650)* mutant which showed the most deficient pumping phenotype (Fig 3) did not show a

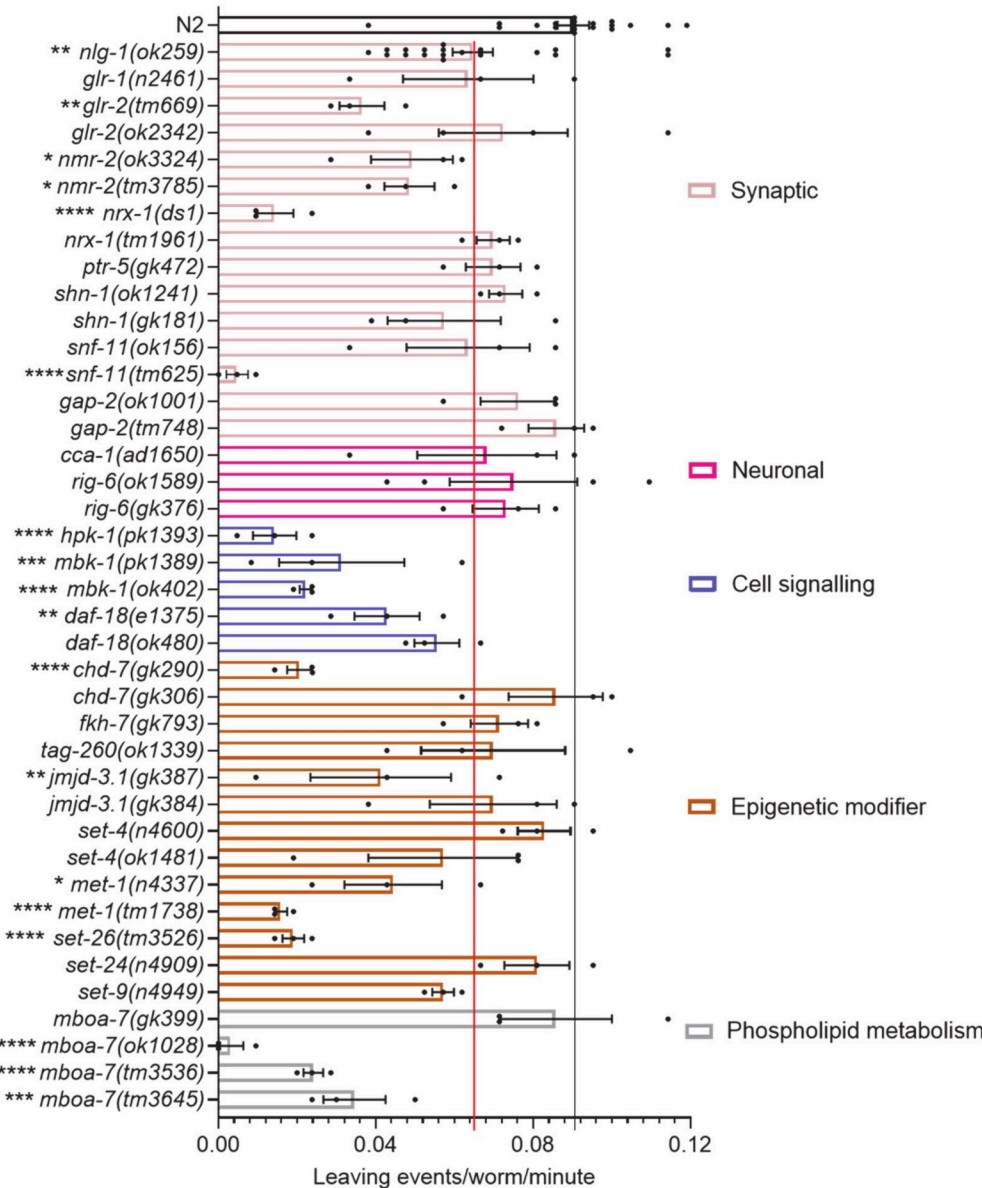

**Fig 2. Food leaving behaviour of *C. elegans* mutants after 24 hours on food to investigate human ASD associated genes.** A food leaving assay was performed with N2, *nlg-1(ok259)* and 39 other *C. elegans* mutants. Genes are categorised and colour coded into different functional domains. The black line indicates the number of leaving events/worm/minute for N2 control. The red line indicates the food leaving rate of *nlg-1(ok259)* control. N2 and *nlg-1(ok259)* n = 19. All other mutants n = 3–4, where n refers to the number of replicates of an individual experiment. Strains were screened in batches across different days. Each batch consisted of 4–6 mutant strains and a paired wild-type control. Data plotted includes all wild-type controls. All data shown as mean ±SEM. Statistical analysis performed using a one-way ANOVA and Dunnetts's multiple comparison test; ns, p>0.05; *, p<0.05; **, p p≤0.01; ***, p≤0.001; ****, p≤0.0001. All significance relates to a comparison with N2 control.

food leaving phenotype (Fig 2), further suggesting that deficits in feeding behaviour are unlikely to explain differences in food leaving behaviour.

Next, we measured early development by quantifying the proportion of total progeny that were eggs, L1 and L2 progeny 24 hours after introducing 7 L4+1. We used % proportion to normalise for observed variation in the total number of eggs laid. 75% of mutants developed at

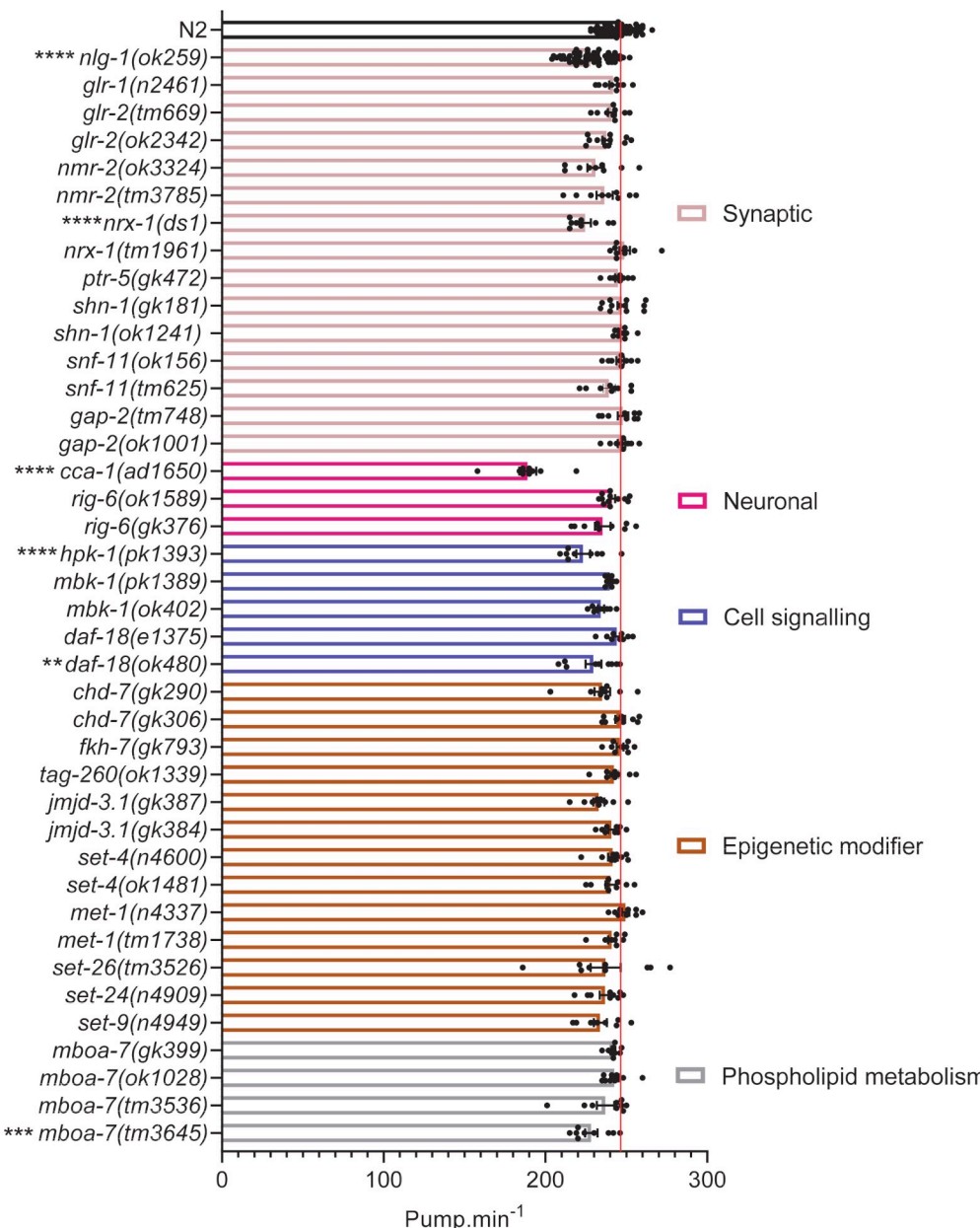

**Fig 3. Pharyngeal pump rate for *C. elegans* mutants.** After a food leaving assay at 24 hours, three worms were chosen at random and their pharyngeal pump rate was counted per minute. N2 and *nlg-1(ok259)* n = 57. All other mutants n = 9–12, where n refers to the number of individual animals investigated. Strains were screened in batches across different days. Each batch consisted of 4–6 mutant strains and a paired wild-type control. Data plotted includes all wild-type controls. The red line indicates pumps per minute for N2 control. All data shown as mean ±SEM. Statistical analysis performed using a one-way ANOVA and Dunnetts's multiple comparison test; ns, $p > 0.05$; *, $p < 0.05$; **, $p \leq 0.01$; ***, $p \leq 0.001$; ****, $p \leq 0.0001$. All significance relates to a comparison with N2 control.

a similar rate to N2 showing that there is no gross early developmental delay (Fig 4). Interestingly, whilst early development seems to be largely unaffected we noted a larger variation in the egg laying of distinct mutants when compared to N2 controls (S1 Fig). The number of eggs laid by a mutant is an important consideration for this assay because the density of progeny populating a food lawn is known to influence the food leaving rate of adult worms [38]. We

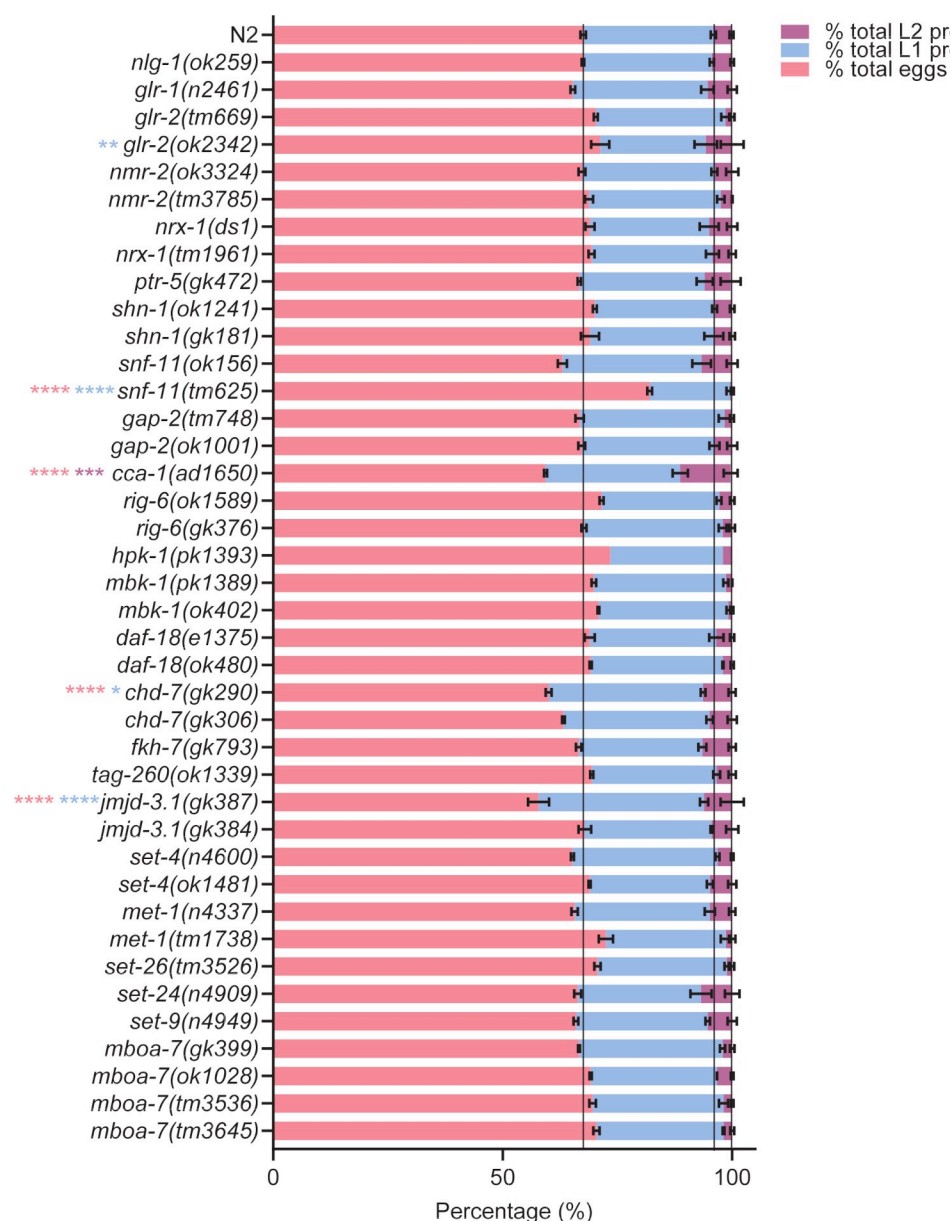

**Fig 4. Percent total eggs and progeny produced by *C. elegans* mutants at 24 hours.** After a food leaving assay from naïve food lawns occupied by 7 L4+1, the percent total offspring that were eggs, L1 and L2 progeny were quantified. N2 and *nlg-1(ok259)* n = 19. All other mutants n = 3–4, where n refers to the number of replicates of an individual experiment. Strains were screened in batches across different days. Each batch consisted of 4–6 mutant strains and a paired wild-type control. Data plotted includes all wild-type controls. The black lines indicate % total eggs, % total L1 progeny and % total L2 progeny for N2 control. Pink asterisks indicate statistical difference between mutant and N2 for % total eggs. Blue asterisks indicate statistical difference between mutant and N2 for % total L1 progeny. Purple asterisks indicate statistical difference between mutant and N2 for % total L2 progeny. All data shown as mean ±SEM. Statistical analysis performed using a two-way ANOVA and Tukey's multiple comparison test; ns, p>0.05; *, p<0.05; **, p≤0.01; ***, p≤0.001; ****, p≤0.0001.

plotted the relationship between the number of progeny produced by a mutant and the food leaving behaviour and showed that the two were correlated (Fig 5). Interestingly, this applies to the *nrx-1* and *chd-7* mutants for which the two alleles tested resulted in distinct social phenotypes (Fig 2). In each case the mutant that showed impaired social behaviour (Fig 2) also

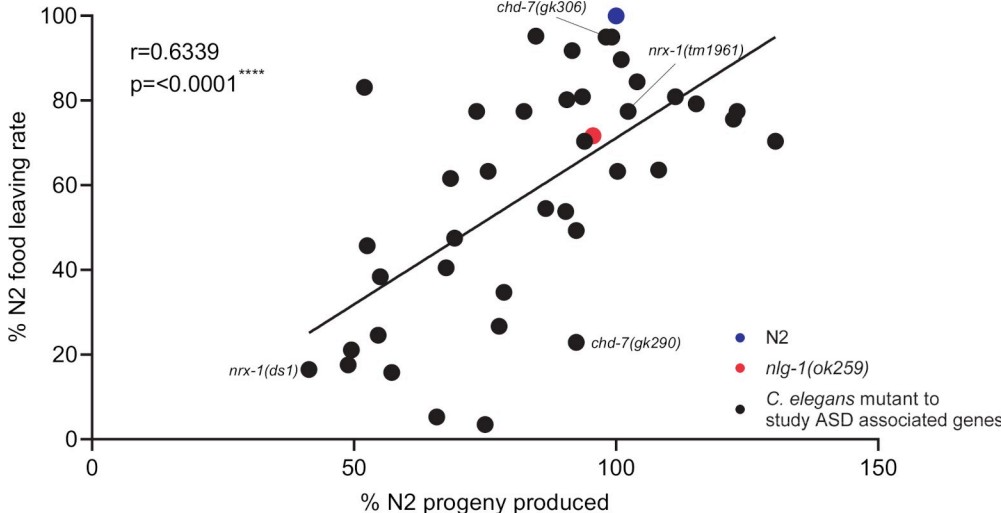

**Fig 5. Correlation between food leaving behaviour of *C. elegans* mutants and progeny production during food leaving assay.** The percent food leaving rate and progeny produced for each *C. elegans* mutant was calculated in comparison to N2. N2 and *nlg-1(ok259)* n = 19. All other mutants n = 3–4, where n refers to the number of replicates of an individual experiment. All data shown as mean. Statistical analysis performed using Pearson correlation coefficient.

produced fewer progeny (S1 Fig). Producing fewer progeny means adult worms were exposed to fewer progeny-derived social cues [38] and could explain the low food leaving rate seen. The correlation between social behaviour and progeny exposure, and the limited disruption seen to the other phenotypes tested, implies that progeny-derived social cues selectively effect social behaviour and therefore progeny exposure is an important consideration in this type of investigation.

### Exposure to N2 progeny selectively modulates social behaviour in a number of mutants

Those strains which produced few progeny confound the assessment of the reduced food leaving behaviour as the response is dependent on the density of progeny populating the food lawn [38]. This was addressed by testing the food leaving behaviour of 30 mutants in response to an experimentally controlled number of N2 progeny. This used a pre-conditioning approach in which N2 progeny pre-populate the lawn and precondition them by mimicking the progeny population that emerge in the first food leaving assay. These assays allow the acute effect of progeny exposure on food leaving behaviour to be investigated. This secondary screen focussed on mutants that showed a mean food leaving rate lower than that of *nlg-1(ok259)* in at least one allele tested (Fig 2). Thus we directly tested the veracity of mutants that emerge from the first screen and explicitly address the potential confound of reduced progeny number. For each mutant we performed a paired experiment in which mutant food leaving was measured on a naïve, unmatched control, plate containing OP50 and a preconditioned plate that incubated 140–200 eggs for 24 hours before introducing 7 L4+1 adults. In accordance with previous findings, N2 adults showed enhanced food leaving in response to progeny and this response was blunted in the *nlg-1(ok259)* adults exposed to pre-loaded N2 progeny (Fig 6).

Analysis of mutants in response to pre-loaded N2 progeny revealed a number of mutants which left infrequently on both naïve and pre-conditioned food lawns, showing little progeny-enhanced food leaving (Fig 6). We reasoned that the low food leaving rate of these mutants

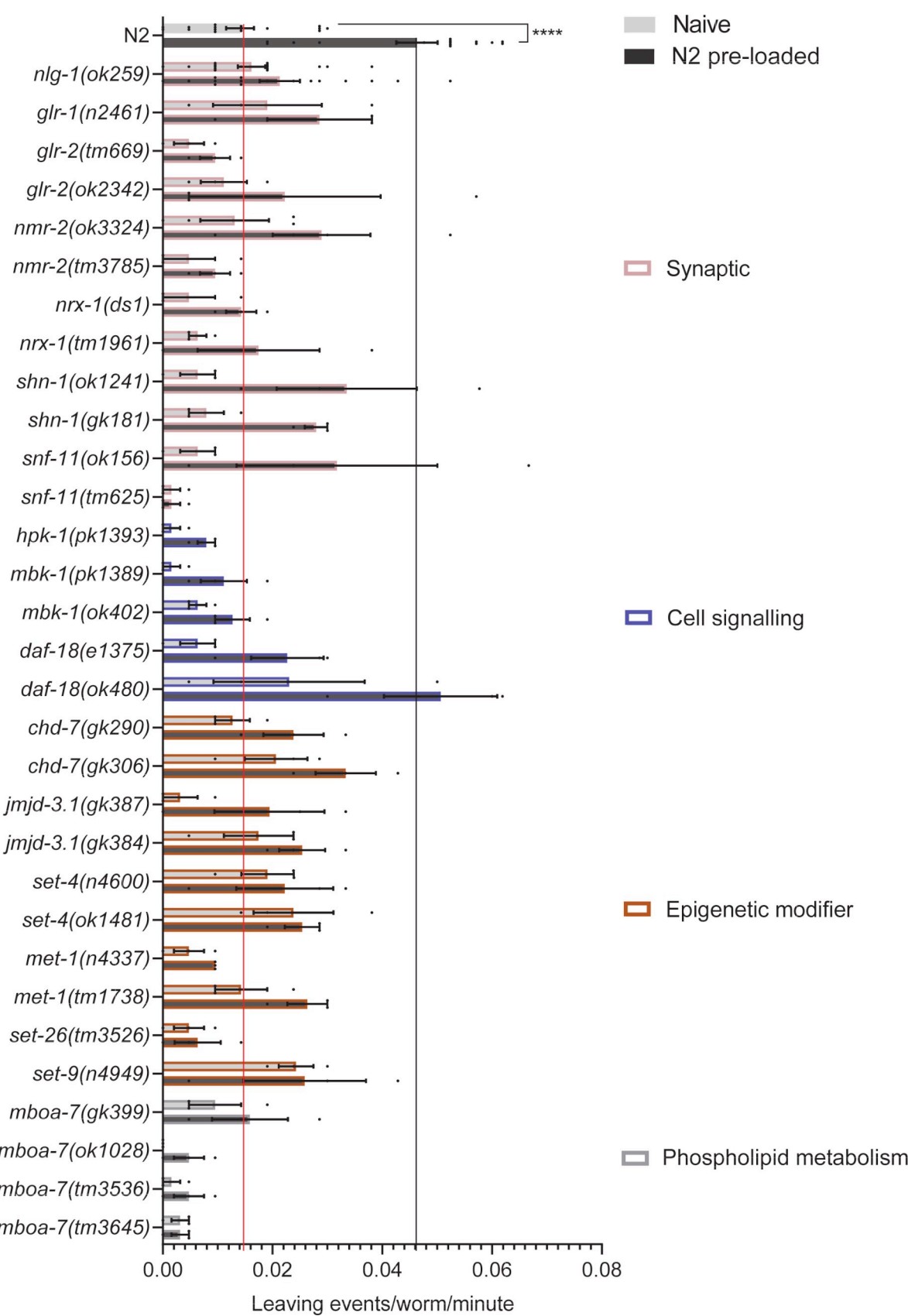

**Fig 6. Food leaving behaviour of *C. elegans* mutants in the absence of progeny and exposure to N2 progeny.** A food leaving assay was performed with N2, *nlg-1(ok259)* and 30 other *C. elegans* mutants on naïve and pre-conditioned food lawns. A naïve lawn contains no progeny whereas a pre-conditioned food lawn contains ~150–200 N2 progeny. The red line indicates the food leaving rate of N2 naïve control. The black line indicates the food leaving rate of the N2 pre-loaded control. Data shown as mean ±SEM. N2 and *nlg-1(ok259)* n = 16. All other mutants n = 3–4, where n refers to the number of replicates of an individual experiment. Strains were screened in batches across different days. Each batch consisted of 4–6 mutant strains and a paired wild-type control. Data plotted includes all wild-type controls. Statistical analysis performed using a two-way ANOVA and Sidak's multiple comparison test; ns, $p > 0.05$; $p \leq 0.001$****.

could be explained by locomotory deficits. To address this we performed a thrashing assay to assess the innate movement ability of the mutants. 12 of the 31 mutants tested showed minor disruption to thrashing behaviour (Fig 7A). For the majority of mutants thrashing did not predict food leaving phenotype. For example, *nlg-1(ok259)* and *set-4* mutants showed deficits in food leaving behaviour (Fig 6) without any impairment to thrashing (Fig 7A). Furthermore, four mutants of the *mboa-7* gene all showed impaired progeny-induced food leaving behaviour with only one of these mutants having reduced thrashing (Fig 7A). The bioinformatic pipeline filtered out mutant strains deficient in gross motility. In addition, we investigated the association between thrashing and food leaving behaviour and identified that there is a correlation between these behaviours (Fig 7B). This suggests that, whilst there are examples of mutants that have impaired food leaving in the absence of a thrashing phenotype, we cannot discount more subtle motility defects contributing to the summed food leaving response. The social impairment we observed in mutants suggests that a variety of genes may act as molecular determinants of social behaviour. Interestingly, these genes were part of synaptic, cell signalling, epigenetic modifier and phospholipid metabolism categories. This highlights that molecular determinants from these biological domains may be important for the emergence of social behaviour.

The comparison of behaviour on naïve and pre-conditioned lawns also allowed for the analysis of egg laying behaviour in response to progeny. We quantified the number of eggs laid by each mutant on naive and pre-conditioned food lawns after each food leaving assay. Interestingly, all mutants laid the same number of eggs on naïve and pre-conditioned food lawns (S2 Fig). This shows that progeny exposure modulates food leaving behaviour and not egg laying. This therefore suggests that the circuit which integrates progeny cues to sculpt food leaving motility is independent of egg laying behaviour which is modulated by other environmental cues.

Overall, starting with 91 human ASD associated genes we used criteria based filtering to define 21 candidate genes for analysis using a *C. elegans* social paradigm. An initial screen of social behaviour in response to progeny produced over 24 hours indicated 23 mutants with a reduced food leaving phenotype. We then confirmed the veracity of this phenotype using a progeny pre-conditioned food leaving approach and identified mutants that showed a socially impaired phenotype. The limited disruption in other phenotypes tested for these mutants suggests that reduced food leaving is a selective social impairment in response to progeny-derived social cues. Identification of these mutants highlights genetic determinants that appear to play a role in social behaviour and also suggests that a number of biological domains (synaptic, cell signalling, epigenetic modification and phospholipid metabolism) are important for the coordination of social behaviour.

## Discussion

ASD is characterised by a triad of behavioural impairments including neuro-atypical behaviour in the social domain [1]. Individuals with ASD have also been shown to produce altered behavioural responses to a range of chemosensory cues such as olfactory, tactile and gustatory

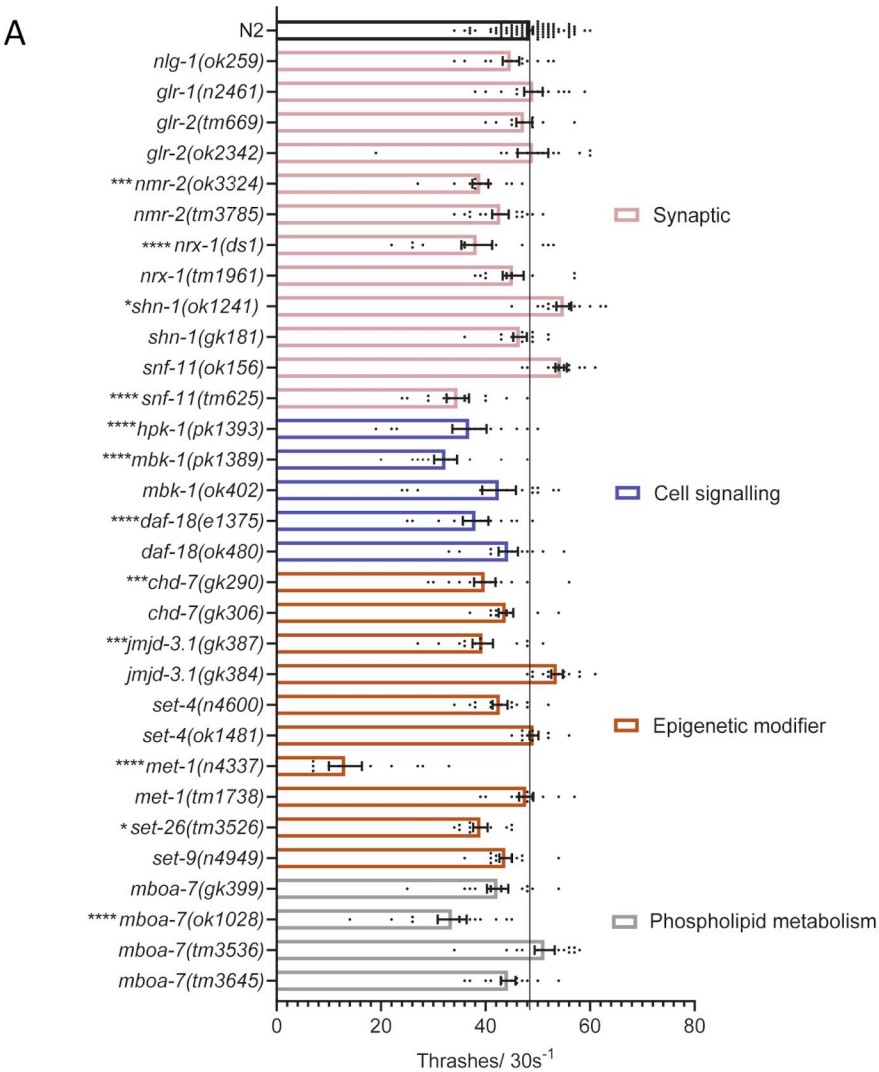

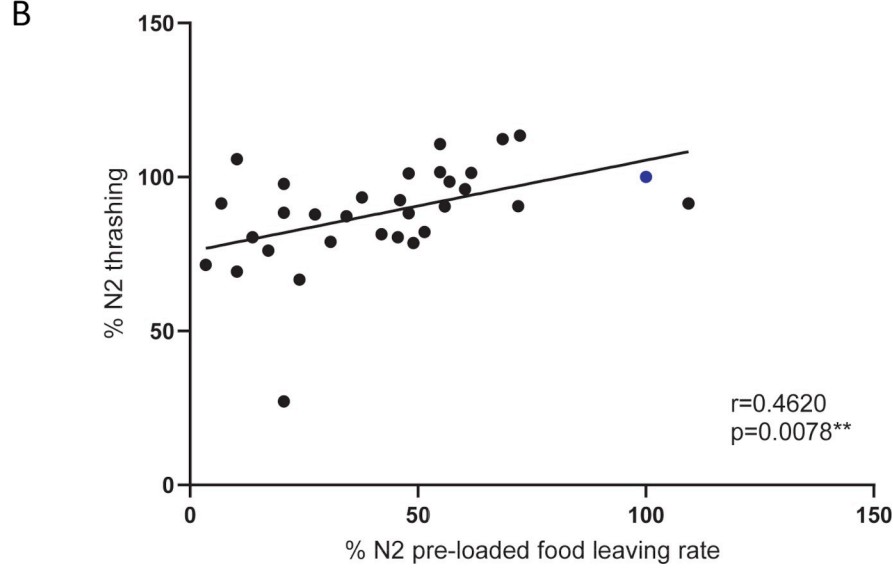

**Fig 7. Thrashing behaviour of *C. elegans* mutants compared to N2.** A 5–10 minutes after being picked into liquid medium, *C. elegans* thrashing behaviour was measured for 30 seconds per worm. The black line indicates the thrashes/30s of N2 control. All data shown as mean ±SEM. Statistical analysis performed using a one-way ANOVA and Dunnetts's multiple comparison test; ns, $p > 0.05$; *, $p < 0.05$; ***, $p \leq 0.001$; ****, $p \leq 0.0001$. All significance relates to a comparison with N2 control. B Correlation between thrashing (Fig 7A) and food leaving behaviour on N2 pre-loaded food lawns (Fig 6). The percent thrashing rate and food leaving rate on N2 pre-loaded food lawns for *C. elegans* mutants was calculated in comparison to N2. N2 is indicated by the datapoint coloured in blue. All data shown as mean. Statistical analysis performed using Pearson correlation coefficient. For thrashing experiments N2 n = 88. All other mutants n = 10–13, where n refers to the number of individual animals investigated. Strains were screened in batches across different days. Each batch consisted of 3–8 mutant strains and a paired wild-type control. For food leaving assays N2 and *nlg-1(ok259)* n = 16. All other mutants n = 3–4, where n refers to the number of replicates of an individual experiment. Strains were screened in batches across different days. Each batch consisted of 4–6 mutant strains and a paired wild-type control.

cues [5,83]. Multi-sensory processing deficits identified in ASD highlights the importance of sensory integration at a circuit level [84] however, it is still unclear how disruption within neural circuits evoke a modified behavioural output. Recent experiments have highlighted the value of investigating molecular determinants of ASD in the context of defined integrative circuits to try and understand more precisely how disruption within these circuits underpins the phenotypes associated with ASD [16,17,36,47]. Approaches, such as these, that better resolve the underlying mechanisms should facilitate pharmacological treatment of ASD and other neuropsychiatric disorders [85].

ASD is known to have a complex genetic architecture, with hundreds of genes with varying penetrance implicated in its aetiology [7]. Although the genetic basis is well documented the functional contribution which many of the genes make to the behavioural domains associated with ASD is unclear. Animal models have begun to understand genetic contribution in autism [26]. The analysis of single, high penetrant, variants is becoming increasingly well refined with the use of animal social behaviours. Use of social behaviours underpinned by discrete neural circuits has helped establish the role of some ASD associated genes in the social domain [39,86]. However, the analysis of common, low penetrant, variants is more complex. Additive effects from polygenic interaction of multiple common variants contributes to wide spread disruption at distinct levels of the biological system which is expressed as an emergent behaviour [8].

*C. elegans* have been used in targeted single gene approaches and in screens of ASD associated genes to provide valuable insight into the role of some genes in sensory processing, development and learning phenotypes [36,87]. Recently we have shown the utility of using a social behavioural paradigm in *C. elegans* to investigate a single ASD associated gene [39]. This paradigm is based on inter-organismal signalling by use of chemosensory social cues which results in a progeny-induced food leaving phenotype [38]. In this study we have used this social paradigm in a screen of ASD associated genes and identified gene candidates with a role in *C. elegans* social behaviour.

We created a pipeline to prioritise human genes for investigation using *C. elegans* social behaviour. The pipeline identified 84% of the ASD genes ranked in categories 1 and 2 by SFARI have an orthologue in *C. elegans*. In general, the rate of conservation between the human and *C. elegans* genomes is 53% [88] and suggests an enriched conservation of ASD genes between humans and *C. elegans*. A large proportion of the orthologous genes had known developmental phenotypes and were not selected for further analysis in this study. The analysis of such genes has the potential to provide broader insight into other neuropsychiatric disorders, such as schizophrenia given the overlap of risk genes between this disorder and ASD and the shared neurodevelopmental component of these two conditions. We selected 21 human genes for investigation using 40 *C. elegans* mutant orthologues. Similarities between

our prioritisation strategy and those used in previous *C. elegans* studies resulted in the iterative selection of some well-studied ASD associated genes such as neuroligin and neurexin [34,36]. Our study is distinct from others because we biased our gene filtering approach to select for *C. elegans* mutants that were appropriate for analysis of social behaviour using a pre-conditioned food leaving approach. We were selective in choosing mutants appropriate for our behavioural analysis, for example the exclusion of overt locomotory mutants due to their possible confounding effect on food leaving motility.

We used a single point analysis focused on progeny induced food leaving from which we also analysed pharyngeal pumping, early development and egg laying capabilities in response to progeny derived social cues. We identified a number of mutants with an altered behavioural response to progeny populating a food lawn providing evidence that *C. elegans* are capable of modelling disruption to an emergent behaviour in response to mutation to an ASD associated gene. Where possible more than a single variant for a particular gene of interest was tested. Whilst in the majority of cases there was concordance in the behavioural output of variants, there were some exceptions and this may reflect differences in the number of times strains were outcrossed and potential differences in genetic backgrounds. Refined parametrization of the sub-behaviour that underpins the food leaving behaviour, combined with precise genetic lesion will facilitate the systematic testing of alleles with equivalence to human mutations [39].

Movement in liquid has been used in other studies to screen ASD associated genes [34]. Our analysis of thrashing in mutants identified that this type of locomotory assay does not accurately predict an impaired food leaving behaviour and does not serve as a surrogate for the more complex integrative progeny-induced social behaviour phenotype. This highlights our behavioural screen as a unique platform which is selectively tuned to identify genetic determinants with a role in social circuits, that when disrupted could appear phenotypically normal in thrashing behaviour.

The candidates that we identified as having a role in social behaviour are orthologues of human genes that range in function including synaptic, cell signalling and epigenetic modification. Genes disrupted in each of these domains are known to contribute to ASD [89,90]. Therefore, the genes that emerge from our screen are representative of the main functional domains disrupted in autism. Our screen has therefore produced a diverse list of candidate genes that can be used to interrogate the systems level disruption that evokes modified behavioural output in ASD. Previous work has focused largely on locomotory and morphological readouts of altered behavioural phenotypes [34–36] whereas our approach facilitates the identification of candidate genes with a role in a more complex, sensory regulated, emergent behaviour which more closely resembles the social domain disrupted in autism. Identification of candidates using this approach therefore provides a benchmark from which the social circuit can be further dissected [91].

We identified five synaptic genes, *nlg-1*, *nrx-1*, *shn-1*, *glr-1* and *nmr-2*, with a role in coordinating progeny-induced social behaviour. In the mammalian nervous system, NLGN, NRXN and SHANK's interaction at the synapse is well established and dysfunction to all three genes has been widely implicated in ASD [92]. Synaptic scaffolds including SHANK are known to interact with receptors such as AMPA and NMDA to help regulate the ion channel composition at the synapse [93]. This provides evidence that this assay for social interaction identifies behavioural disruption in orthologues of genes that function together at mammalian synapses. The role of these mammalian genes in nervous system function and/or synaptic transmission are functionally conserved in *C. elegans* [40,49,51,94–96]. This means that we can resolve singular determinants with the potential to unpick genes that encode dysfunctional interactions. This raises the opportunity to model the polygenic nature of ASD [8,36,97–99].

Previous scaled use of assays to investigate ASD associated genes in *C. elegans* used strategies to prioritise genes before behavioural analysis [34–36]. The outcome of these studies resulted in an incomplete overlap of some genes investigated in our study. We made a comparison between *C. elegans* mutants that emerged from our study with an impaired social phenotype to mutants that have emerged from previous studies as having impaired movement and habituation phenotypes [34,36]. However, the vast majority of mutants that we identified with behavioural impairment are unique to this study. For example, *shn-1(gk181)* and *set-9(n4949)* show impaired social behaviour in response to progeny, whilst appearing grossly wild-type for the other phenotypes we tested. In addition these mutants do not show a behavioural phenotype in movement or habituation behaviour when investigated in previous studies [34,36]. This highlights that the emergent behaviour we have used reveals genes that are missed when they emerge from the bioinformatic pipeline. This makes the case that applying a lower throughput observer based assay will refine previous efforts to model the functional impact of genes implicated in ASD.

The emergent behaviour that we have used is a complex, sensory integrative behaviour. Habituation learning is another complex behaviour in *C. elegans* that has been investigated in a previous screen of ASD associated genes [36]. Therefore, we wanted to identify whether there was overlap in mutants with behavioural impairment in two distinct complex behavioural phenotypes. We made a comparison of mutants that we had identified as having a social impairment to mutants that have been shown to have a habituation phenotype [36]. We identified four synaptic mutants, *nlg-1(ok259)*, *nrx-1(ds1)*, *glr-1(n2461)* and *nmr-2(ok3324)*, which have impaired social behaviour and have also been shown to have a habituation phenotype. This suggests that these genes may have a role in coordinating more than one complex sensory-regulated behaviour in *C. elegans*. This may also suggest that a key role of synaptic genes is in coordinating higher behaviours in *C. elegans*. With this in mind it would be interesting to extend the analysis of mutants with habituation impairment and screen them for social deficits. Our approach lends itself to the identification of complex behavioural deficits and so would be valuable in this analysis to further understand if there is an over-representation of synaptic genes in complex sensory integrative phenotypes.

In addition to highlighting the important contribution of synaptic, cell signalling and epigenetic genes, we identify a gene involved in phospholipid metabolism that appears to play a role in progeny-induced social behaviour. *mboa-7* is a lysoPI acetyltransferase which is important in the regulation of phospholipid membranes [100] and cell signalling [78] in *C. elegans*. In mammals, the regulation of membrane composition is important for cellular processes, signalling and nervous system function [101,102]. Studies of the MBOAT7 ortholog in mice suggest it may function in brain development [103], however this gene is comparatively less well studied than other ASD associated genes for its functional contribution to the disorder. Therefore, the identification of this gene with a role in progeny-induced social behaviour highlights how this study enriches the understanding of the molecular determinants of social behaviour from underrepresented genes in autism.

In conclusion, investigation of ASD associated orthologues in *C. elegans* identified genes from a number of candidates implicated in ASD that disrupt social behaviour in the worm. Identification of these genes highlights how this assay might be used in quantitative approaches that can probe the single [39] and polygenic nature of ASD and its underpinning genetic architecture [36]. The robust nature of this assay provokes a better detailing of the cellular and circuit dependence of this social interaction. Guided by the cellular determinants of behaviour, investigation can extend to probe the polygenic nature of ASD and take a similar approach in other psychiatric diseases that have significant consequences for behavioural traits in the social domain [104,105].

## Supporting information

**S1 Fig. Proportion of eggs and progeny produced by *C. elegans* mutants after 24 hours.**
After a food leaving assay the number of eggs and progeny produced after 24 hours was
counted. N2 and *nlg-1(ok259)* n = 19. All other mutants n = 3–4, where n refers to the number
of replicates of an individual experiment. Strains were screened in batches across different
days. Each batch consisted of 4–6 mutant strains and a paired wild-type control. Data plotted
includes all wild-type controls. The black line indicates the total number of eggs and progeny
produced by N2 control. All data shown as mean ±SEM. Statistical analysis performed using a
two-way ANOVA and Dunnetts's multiple comparison test; ns, $p > 0.05$; *, $p < 0.05$; **, $p \leq 0.01$;
***, $p \leq 0.001$; ****, $p \leq 0.0001$. All significance relates to the total number of eggs and progeny
produced in comparison with N2 control.
(TIF)

**S2 Fig. The number of eggs laid is unchanged in the presence of progeny.** After a food leav-
ing assay the number of eggs laid was quantified. The black line indicates the number of eggs
laid by N2 control. All data shown as mean ±SEM. N2 and *nlg-1(ok259)* n = 16. All other
mutants n = 3–4, where n refers to the number of replicates of an individual experiment.
Strains were screened in batches across different days. Each batch consisted of 4–6 mutant
strains and a paired wild-type control. Data plotted includes all wild-type controls. Statistical
analysis performed using a two-way ANOVA and sidak's multiple comparison test. No signifi-
cance was identified when comparing eggs laid on a naïve and pre-loaded food lawn for each
strain.
(TIF)

## Acknowledgments

Some strains were provided by the CGC. Some strains were provided by the National BioRe-
source Project (NBRP) Japan.

## Author Contributions

**Conceptualization:** Helena Rawsthorne, Fernando Calahorro, Lindy Holden-Dye, Vincent O'
Connor, James Dillon.

**Data curation:** Helena Rawsthorne, Fernando Calahorro, Lindy Holden-Dye, Vincent O'
Connor, James Dillon.

**Formal analysis:** Helena Rawsthorne, Fernando Calahorro, Lindy Holden-Dye, Vincent O'
Connor, James Dillon.

**Funding acquisition:** Lindy Holden-Dye, Vincent O' Connor, James Dillon.

**Investigation:** Helena Rawsthorne, Fernando Calahorro, Lindy Holden-Dye, Vincent O' Con-
nor, James Dillon.

**Methodology:** Helena Rawsthorne, Fernando Calahorro, Lindy Holden-Dye, Vincent O' Con-
nor, James Dillon.

**Project administration:** Helena Rawsthorne, Fernando Calahorro, Lindy Holden-Dye, Vin-
cent O' Connor, James Dillon.

**Resources:** Lindy Holden-Dye, Vincent O' Connor, James Dillon.

**Supervision:** Lindy Holden-Dye, Vincent O' Connor, James Dillon.

**Validation:** Helena Rawsthorne, Fernando Calahorro, Lindy Holden-Dye, Vincent O'
   Connor.

**Visualization:** James Dillon.

**Writing – original draft:** Helena Rawsthorne, Fernando Calahorro, Lindy Holden-Dye, Vin-
   cent O' Connor, James Dillon.

**Writing – review & editing:** Helena Rawsthorne, Fernando Calahorro, Lindy Holden-Dye,
   Vincent O' Connor, James Dillon.

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
