## [Decision Letter · Decision Letter 0]

13 Jan 2021

PONE-D-20-37216

Investigating autism associated genes in C. elegans reveals candidates with a role in social behaviour

PLOS ONE

Dear Dr. Dillon,

Thank you for submitting your manuscript to PLOS ONE. After careful consideration, we feel that it has merit but does not fully meet PLOS ONE’s publication criteria as it currently stands. Therefore, we invite you to submit a revised version of the manuscript that addresses the points raised during the review process.

Particularly, methods must be clarified and conclusions must be revised to support the real results. The reviewers have provided specific suggestions on data analysis and interpretation that will highly help address these points.

We look forward to receiving your revised manuscript.

Kind regards,

Inmaculada Riquelme

Academic Editor

PLOS ONE

Journal Requirements:

Reviewers' comments:

Reviewer's Responses to Questions

**Comments to the Author**

1. Is the manuscript technically sound, and do the data support the conclusions?

Reviewer #1: No

Reviewer #2: Yes

2. Has the statistical analysis been performed appropriately and rigorously? 

Reviewer #1: No

Reviewer #2: Yes

3. Have the authors made all data underlying the findings in their manuscript fully available?

Reviewer #1: Yes

Reviewer #2: Yes

4. Is the manuscript presented in an intelligible fashion and written in standard English?

Reviewer #1: Yes

Reviewer #2: Yes

5. Review Comments to the Author

Reviewer #1: The manuscript by Rawsthorne and colleagues makes use of the C. elegans model system to investigate genes that have been associated with autism spectrum disorder. As autism is associated with complex behaviours, it is often wondered if simpler model organisms can be used to study genes linked to autism, and importantly to investigate behavioural phenotypes. Here the authors turn to a list of genes curated by SFARI to assemble a collection of C. elegans mutants, that contain genes orthologous to ASD genes, and use them in a variety of assays. The concept is interesting and could be a great system to advance ASD studies. However, some aspects of the study need to be addressed, thus the claims by the authors are not fully supported. Additional clarification, and analyses are required before considering for publication.

It seems to me that the food leaving assay may also be measuring the ability of the worms to move. If the animals have motility problems, then a great number of behavioral phenotypes are likely to be affected. The authors state they did not select strains that showed lethal, sterile or uncoordinated phenotypes, which is reasonable for the behavioural assays they wish to perform.

However, I feel that of their remaining candidate genes, they should have tested whether the mutants displayed uncoordinated phenotypes prior to using them for the food leaving assay.

Re: pharyngeal pumping

238. This shows that most mutants with reduced food leaving behaviour are capable of responding to food‐dependent sensory cues and co‐ordinating normal feeding behaviour.

Does the pharynx not continuously pump as it is largely an independent nervous system, thus I do not think this statement is accurate.

238. In addition, the cca‐1(ad1650) mutant which showed the most deficient pumping phenotype (Fig 3) did not show a food leaving phenotype (Fig 2), further suggesting that deficits in feeding behaviour are unlikely to explain differences in food leaving behaviour.

I do not think they can draw this conclusion from this observation.

238. We plotted the relationship between the number of progeny produced by a mutant and the food leaving behaviour and showed that the two were correlated (Fig 5). Interestingly, this applies to the nrx‐1 and chd‐7 mutants for which the two alleles tested resulted in distinct social phenotypes (Fig 2). In each case the mutant that showed impaired social behaviour (Fig 2) also produced fewer progeny (S1 Fig). Producing fewer progeny means adult worms were exposed to fewer progeny‐derived social cues and could explain the low food leaving rate seen.

I really do not think they can draw these conclusions from these correlations. First, they have not measured (or identified) these progeny-derived social cues, and second these phenotypes could simply arise from overall impaired motility. In this aspect, it is essential to investigate general movement vs. food leaving behaviors.

I realize that the thrashing assay was conducted on these mutants, but I do not think this was done using automated measures (was this done manually?), and thrashing is an acute swimming response that does not fully recapitulate movement on plates during aging. Also, the authors should try to correlate the thrashing data to the food leaving phenotype, as they did for progeny vs. food leaving in Figure 5.

263. Whilst some mutants showed minor disruption to thrashing behaviour (Fig 7) for most mutants thrashing did not predict food leaving behaviour. For example, nlg‐1(ok259) and set‐4 mutants showed deficits in food leaving behaviour (Fig 6) without any impairment to thrashing (Fig 7). Furthermore, four mutants of the mboa‐7 gene all showed impaired progeny‐induced food leaving behaviour with only one of these mutants having reduced thrashing (Fig 7).

I do not think this is accurate. Looking at the figure around 30% of the mutants showed thrashing rates less than N2. This section should be rewritten to reflect the reality of the phenotypes. Also, these data should be directly compared to each mutants corresponding food leaving phenotype.

Perhaps I am reading this incorrectly, but there seems to be a huge discrepancy between the number of control worms and mutants tested for multiple experiments (figures 6,7, Supplementary figure 2 for example). One example is 88 N2 worms and 10-13 mutants in Fig. 7.

Reviewer #2: Hawsthorne et al. set out to examine possible connections between genes implicated as risk factors for autism spectrum disorders (ASD) and measurable phenotypes in C. elegans. In particular, they were interested in the effects of mutations in relevant genes on social behavior manifested as food leaving in response to progeny density. They developed a pipeline for providing candidate genes for examination in behavioral assays. In this paper, they report that a number of genes implicated in ASD produce altered social behavior, but tend to not affect wider dimensions of behavior such as egg laying. The findings are tantalizing, the studies were well designed and the results merit publication in my view. The paper might be strengthened by addressing the comments below.

1. The methods were slightly unclear to me. In describing the Food Leaving Assay on p. 11, the authors mention animals were incubated on the plates for 24 hours (line 137) before assessing food leaving. Then in the next section (Pre-conditioned Food Leaving Assay), they mention that the plates were incubated for 2 hours (line 155) before leaving events were observed. I think this may be a typo, but if not, they need to justify the reason for the time differential in the plate incubation step.

2. At the start of the Results section, the authors observe that 84% of the class 1/2 ASD genes selected for this study have an orthologue in C. elegans. This appears to be a high rate of conservation of ASD genes between humans and C. elegans and is higher than the rate of conservation of genes in general or throughout these two genomes. Plus, a large proportion of the ASD risk genes appear to be essential genes causing lethality or sterility. This merits further comment as it fits with observations of conservation of risk genes in other neuropsychiatric disorders such as schizophrenia, which also has a strong neurodevelopmental component.

3. The reference to ‘n’ in the Figure legends needs some further clarification. In some cases, e.g., Fig. 3, I think ‘n’ refers to the number of individual animals examined for those data. However, in other cases, e.g., Figs. 2, 4 & 5, ‘n’ appears to refer to the number of replicates of an individual experiment. If this interpretation is correct, it would help to explicitly state this to avoid confusion.

4, The authors did a commendable job of testing more than a single variant strain related to a particular gene, where possible. In many cases, there was concordance in the behavior of the different strains in the behavioral assays. There were a few notable exceptions that merit further discussion or exploration.

a) The nrx-1(ds1) allele suppressed food leaving whereas nrx-1(tm1961) did not. The corresponding SG1 strain has been outcrossed 3 times, whereas the FXO1961 strain has not been outcrossed. So part of the difference between the two strains could be due to background gene variation. This could be explored or at least merits some discussion. RNAi against nrx-1 extends lifespan. Is the lifespan of either of these strains similarly affected? The nrx-1(tm1961) allele (truncated protein, presumably less functional) is associated with changes in enhanced slowing response (Rodriguez-Ramos et al. [2017] Behav Genet 47:596). Can this assay be used to probe the differences between these two strains? Finally, another strain VC1416 nrx-1(ok1649) is available potentially as a tiebreaker of sorts. These suggested experiments could help unravel what the true phenotype is for nrx-1 loss of function alleles. Further discussion of these issues is at least warranted.

b) The snf-11(tm625) allele suppressed food leaving whereas snf-11(ok156) did not. This situation is perhaps trickier to explain. Both are putative null alleles. RM2710 snf-11(ok156) has been outcrossed 6 times, mostly eliminating any stray variation, so this would be the standard for comparison, by most viewpoints. In this scenario, snf-11 would not be considered to affect social behavior in C. elegans. Background variation in snf-11(tm625), which has not been outcrossed, might then be responsible for the altered behavior. Mutant animals with the snf-11(ok156) allele have been reported to have some minor locomotor phenotypes such as increased sinusoidal amplitude during movement and an increase in forward velocity and roaming (Yemini et al. [2013] Nat Methods 10:877). Can these phenotypes be used to sort out the true role of snf-11 in food leaving behavior? In other words, can they determine whether spurious variants in FXO0625 snf-11(tm625) may cause the altered food leaving? Some of these issues and alternative interpretations need to be included in the Discussion.

Despite the comments in this last section, the effort to pin down contributions has been admirable. It would just help to show some of the limits to our ability to firmly assign cause and effect at this stage.

6. PLOS authors have the option to publish the peer review history of their article (what does this mean?). If published, this will include your full peer review and any attached files.

Reviewer #1: No

Reviewer #2: **Yes: **Donard S Dwyer

---

## [Author Response · Author response to Decision Letter 0]

18 Feb 2021

Please see the document 'Response to Reviewers' which includes figure and addresses reviewer comments.

---

## [Decision Letter · Decision Letter 1]

3 Mar 2021

PONE-D-20-37216R1

Investigating autism associated genes in C. elegans reveals candidates with a role in social behaviour

PLOS ONE

Dear Dr. Dillon,

Thank you for submitting your manuscript to PLOS ONE. After careful consideration, we feel that it has merit but does not fully meet PLOS ONE’s publication criteria as it currently stands. Therefore, we invite you to submit a revised version of the manuscript that addresses the points raised during the review process.

Athought the manuscript has greatly improved with the last review, some minor issues must be still addressed.

We look forward to receiving your revised manuscript.

Kind regards,

Inmaculada Riquelme

Academic Editor

PLOS ONE

Journal Requirements:

Reviewers' comments:

Reviewer's Responses to Questions

**Comments to the Author**

1. If the authors have adequately addressed your comments raised in a previous round of review and you feel that this manuscript is now acceptable for publication, you may indicate that here to bypass the “Comments to the Author” section, enter your conflict of interest statement in the “Confidential to Editor” section, and submit your "Accept" recommendation.

Reviewer #1: (No Response)

Reviewer #2: All comments have been addressed

2. Is the manuscript technically sound, and do the data support the conclusions?

Reviewer #1: Yes

Reviewer #2: Yes

3. Has the statistical analysis been performed appropriately and rigorously? 

Reviewer #1: Yes

Reviewer #2: Yes

4. Have the authors made all data underlying the findings in their manuscript fully available?

Reviewer #1: Yes

Reviewer #2: Yes

5. Is the manuscript presented in an intelligible fashion and written in standard English?

Reviewer #1: Yes

Reviewer #2: Yes

6. Review Comments to the Author

Reviewer #1: In general the authors have done a good job addressing my concerns. There is one minor issue that I would like to be addressed before considering the manuscript for publication.

Line 343 ‘12 of the 31 mutants tested showed minor disruption to thrashing behaviour (Fig 7A). For the majority of mutants thrashing did not predict a food leaving phenotype. For example, nlg‐1(ok259) and set‐4 mutants showed deficits in food leaving behaviour (Fig 6) without any impairment to thrashing (Fig 7A). Furthermore, four mutants of the mboa‐7 gene all showed impaired progeny‐induced food leaving behaviour with only one of these mutants having reduced thrashing (Fig 7A). Consistent with these observations the correlation of thrashing behaviour with food leaving behaviour identified only a weak, positive linear correlation (Fig 7B). Taken together, this suggests that impaired motility may contribute but is not the major determinant of the food leaving behaviour.’

I am happy that the authors did take a second look at potential underlying motility phenotypes that could influence their assays. Afterall, many of these mutations likely affect development and thus it is not surprising that they may manifest phenotypes when assayed as adults. Eliminating animals with gross motility defects makes sense, but animals with less severe effects will likely confound other assays that require movement, namely most C. elegans behavioral assays.

The authors state that since there is a weak (a judgement call here) correlation which they interpret as motility defects not being the major determinant (another judgement call) of food leaving behavior. Instead I think it is more honest to simply state that they cannot discount general motility defects as a contributing factor, and they should say this explicitly.

Reviewer #2: (No Response)

7. PLOS authors have the option to publish the peer review history of their article (what does this mean?). If published, this will include your full peer review and any attached files.

Reviewer #1: No

Reviewer #2: **Yes: **Donard S Dwyer

---

## [Author Response · Author response to Decision Letter 1]

25 Mar 2021

We have addressed the reviewers comments - see document 'Response to Reviewers' and modified the manuscript.

---

## [Editor Report · Decision Letter 2]

29 Mar 2021

Investigating autism associated genes in C. elegans reveals candidates with a role in social behaviour

PONE-D-20-37216R2

Dear Dr. Dillon,

We’re pleased to inform you that your manuscript has been judged scientifically suitable for publication and will be formally accepted for publication once it meets all outstanding technical requirements.

Kind regards,

Inmaculada Riquelme

Academic Editor

PLOS ONE
---

## [Editor Report · Acceptance letter]

18 May 2021

PONE-D-20-37216R2 

Investigating autism associated genes in *C. elegans* reveals candidates with a role in social behaviour 

Dear Dr. Dillon:

I'm pleased to inform you that your manuscript has been deemed suitable for publication in PLOS ONE. Congratulations! Your manuscript is now with our production department. 

Kind regards, 

on behalf of

Dr. Inmaculada Riquelme 

Academic Editor

PLOS ONE